# Less Time, Same Insight? Evaluating Short Functional Tests as Substitutes for the Six-Minute Walk Test and the Reliability and Validity of the 2MWT, 3MWT, and 1MSTS in Bariatric Surgery Candidates with Obesity

**DOI:** 10.3390/healthcare13151883

**Published:** 2025-08-01

**Authors:** Hamdiye Turan, Zeynal Yasaci, Hasan Elkan

**Affiliations:** 1Department of Chest Diseases, Harran University Faculty of Medicine, Sanlıurfa 63290, Türkiye; dr_hamdiyeturan@hotmail.com; 2Department of Physiotherapy and Rehabilitation, Faculty of Health Sciences, Inonu University, Malatya 44000, Türkiye; 3Department of General Surgery, Harran University Faculty of Medicine, Sanlıurfa 63290, Türkiye; dr_elkan@hotmail.com

**Keywords:** obesity, functional capacity, three-minute walk test, two-minute walk test, sit-to-stand, bariatric surgery, aerobic performance

## Abstract

**Background and Objectives:** Functional capacity assessment is essential in bariatric surgery candidates, but the Six-Minute Walk Test (6MWT) may be limited by fatigue, joint pain, and spatial constraints in individuals with severe obesity. Shorter tests such as the Two-Minute Walk Test (2MWT), Three-Minute Walk Test (3MWT), and One-Minute Sit-to-Stand Test (1MSTS) have been proposed as alternatives, yet comparative data in this population remain scarce. We aimed to evaluate the validity, reliability, and clinical utility of the 2MWT, 3MWT, and 1MSTS as substitutes for the 6MWT in patients preparing for bariatric surgery. **Materials and Methods:** In this cross-sectional study, 142 obese adults (BMI ≥ 30 kg/m^2^) underwent standardized 2MWT, 3MWT, 6MWT, and 1MSTS protocols. Correlation, linear regression, test–retest reliability (ICC), and ROC analyses were used to determine each test’s correlation and discriminative accuracy for impaired exercise tolerance (6MWT < 450 m). **Results:** The 3MWT showed the strongest correlation with the 6MWT (r = 0.930) and the highest explained variance (R^2^ = 0.865), especially in individuals with BMI > 50. It also exhibited excellent reliability (ICC > 0.9) and a strong ROC profile (AUC = 0.931; 212 m cut-off). The 2MWT demonstrated acceptable concurrent validity but slightly lower agreement. The 1MSTS showed weak and inconsistent associations with 6MWT performance, suggesting limited value in assessing aerobic capacity in this population. **Conclusions:** The 3MWT appears to be a valid, reliable, and clinically practical alternative to the 6MWT in individuals with severe obesity. The 2MWT may be used when time or patient tolerance is limited. The 1MSTS, while safe and simple, may reflect strength and coordination more than aerobic capacity, limiting its utility in this context.

## 1. Introduction

Obesity is a growing public health challenge worldwide, associated with a wide range of metabolic, cardiovascular, and functional impairments. According to the World Health Organization, as of 2022, approximately 2.5 billion adults were overweight, including 890 million living with obesity [1]. An increased BMI not only elevates cardiometabolic risks but also leads to diminished physical performance, reduced exercise capacity, and impaired daily functioning. These effects are particularly pronounced in bariatric surgery candidates due to reduced skeletal muscle strength, decreased cardiopulmonary capacity, increased gait inefficiency, and the presence of comorbidities such as osteoarthritis and diabetes, which interfere with walking performance in this population [2,3].

Although the cardiopulmonary exercise test (CPET) remains the gold standard in assessing maximal exercise capacity, its use in routine clinical practice is limited due to high costs, technical complexity, and limited accessibility. As a practical alternative, the Six-Minute Walk Test (6MWT) is widely recognized for its ability to reflect submaximal aerobic performance and clinical feasibility [4]. However, in individuals with severe obesity, the 6MWT may be compromised by diminished cardiorespiratory reserves, and its outcomes can be influenced by variables such as age, sex, body morphology, and examiner experience [5,6,7]. Moreover, a notable portion of individuals with BMI > 40 kg/m^2^ may even fail to complete the test due to premature fatigue or musculoskeletal limitations [2].

Consequently, shorter tests such as the Two-Minute Walk Test (2MWT), Three-Minute Walk Test (3MWT), and One-Minute Sit-to-Stand Test (1MSTS) have drawn increased attention due to their simplicity, time efficiency, and lower physical demands. These tests have been individually applied in various populations, including older adults [8], individuals with orthopedic pathologies [9], and people with obesity [10]. However, to our knowledge, no study to date has evaluated the 2MWT, 3MWT, and 1MSTS concurrently within the same obese cohort or directly compared their performance with that of the 6MWT.

Given this gap in the literature, there is a growing need to evaluate short-duration tests that are clinically accessible and capable of providing valid insights into functional capacity. Therefore, this study aimed to assess the test–retest reliability, validity, and clinical utility of the 2MWT, 3MWT, and 1MSTS as alternative field tests to assess functional capacity in individuals preparing for bariatric surgery.

## 2. Methods

### 2.1. Study Design and Participants

This cross-sectional study was conducted at the Department of Chest Diseases and General Surgery Clinic of Harran University Hospital. Eligible individuals were adults aged 18 to 65 years with a body mass index (BMI) ≥ 30 kg/m^2^ and the physical ability to perform submaximal walking and sit-to-stand tests. Patients were excluded if they had severe cardiopulmonary disease, neurological or musculoskeletal limitations, or cognitive impairments that could interfere with test performance. All participants were clinically assessed by a general surgeon and were deemed eligible candidates for bariatric surgery prior to inclusion in the study. The study protocol was approved by the Ethics Committee of Harran University Faculty of Medicine (HRU/25.10.38), and written informed consent was obtained from all participants in accordance with the Declaration of Helsinki.

### 2.2. Power Analysis

The required sample size was estimated based on the correlation-based approach described by Bujang et al. (2016) [11]. Assuming a minimum expected correlation of r = 0.30, a two-tailed significance level of 0.05, and 80% statistical power, the minimum required sample size was calculated as 85 participants [12]. This value reflects a moderate expected correlation based on the prior literature on functional test performance. The final sample of 142 participants was therefore considered adequate for the aims of this study.

### 2.3. Outcome Measures

#### 2.3.1. Demographic and Clinical Characteristics

Participants’ demographic and clinical data were recorded through structured interviews and chart review. These included age, sex, body mass index (BMI), and waist-to-hip ratio (WHR). Smoking status (yes/no) was also noted. BMI was calculated using the standard formula: weight (kg) divided by height squared (m^2^). Waist and hip circumferences were measured using a flexible tape at the midpoint between the last rib and iliac crest (waist) and at the widest point over the buttocks (hip), in a standing position.

#### 2.3.2. Functional Performance Tests

All functional tests were performed by trained physiotherapists in a standardized indoor environment with verbal encouragement and clear instructions. Rest periods were provided between tests to minimize fatigue. To reduce learning and fatigue-related bias, the order of the performance-based tests (1MSTS, 2MWT, 3MWT) was randomized for each participant using a computer-generated randomization list. The tests are detailed below.

*Six-Minute Walk Test (6MWT):* Conducted according to the American Thoracic Society (ATS) guidelines on a 30-m flat corridor. The total distance walked in six minutes was measured in meters. Participants were instructed to walk at a self-paced speed and were allowed to rest if needed [13].

*Three-Minute Walk Test (3MWT) and Two-Minute Walk Test (2MWT):* Performed immediately before the 6MWT using the same corridor and instructions. Distances covered at exactly 2 and 3 min were recorded, while time was tracked using a stopwatch [14,15].

*One-Minute Sit-to-Stand Test (1MSTS):* Participants were seated in a standard-height armless chair (approximately 45 cm seat height). They were instructed to rise to full extension and return to sitting as many times as possible within one minute, without using their arms. The total number of complete repetitions was recorded [10].

All tests were repeated under identical conditions approximately 6 to 8 h after the initial session on the same day, to assess short-term test–retest reliability.

### 2.4. Statistical Analysis

All statistical analyses were conducted using IBM SPSS Statistics version 27.0. Descriptive statistics were presented as the mean ± standard deviation (SD) for continuous variables and as frequencies and percentages for categorical variables.

To explore the association between short-duration functional tests and the 6MWT, Pearson correlation analysis was initially performed. Tests demonstrating the strongest correlations were subsequently evaluated using simple linear regression analysis, with the 6MWT as the dependent variable. The coefficient of determination (R^2^) was reported to indicate the proportion of variance in the 6MWT explained by each short test.

To evaluate the discriminative ability of short tests in identifying individuals with reduced exercise capacity (defined as 6MWT < 450 m), receiver operating characteristic (ROC) curve analysis was performed. The area under the curve (AUC), optimal cut-off values, sensitivity, and specificity were reported. Directional transformation of the test variables was applied where necessary to comply with ROC assumptions. Test–retest reliability for the 3MWT was assessed using the intraclass correlation coefficient (ICC) based on a two-way mixed-effects model, single measure, and absolute agreement definition (ICC(3,1)), as recommended by Koo and Li [16]. The measurement reliability of the 3MWT was evaluated using the intraclass correlation coefficient (ICC), standard error of measurement (SEM), and minimum detectable change (MDC) at 95% confidence. The SEM was calculated using the formula SEM = SD × √(1 − ICC), and the MDC was derived as MDC_95_ = SEM × 1.96 × √2.

## 3. Results

A total of 142 participants (mean age 34.94 ± 10.61 years; 80.3% female) were included in the study. The mean BMI was 45.92 ± 7.09 kg/m^2^, and the average 6MWT distance was 412.37 ± 81.11 m. The mean values for the short functional tests were 206.72 ± 30.75 m for the 3MWT, 140.27 ± 20.02 m for the 2MWT, and 20.98 ± 4.14 repetitions for the One-Minute Sit-to-Stand Test (Table 1).

The 3MWT had the strongest correlation with the 6MWT across all BMI categories, especially in the BMI > 50 group (r = 0.930, *p* < 0.001). The 2MWT also showed strong correlations, whereas the 1MSTS exhibited weaker and inconsistent associations (Table 2).

Simple linear regression analyses demonstrated that the 3MWT explained a substantial portion of the variance in the 6MWT across BMI groups, with the highest R^2^ observed in individuals with BMI > 50 (R^2^ = 0.865, *p* = 0.001). The 2MWT also showed a moderate level of correlation with the 6MWT (Table 3).

The test–retest reliability analysis for the 3MWT yielded an excellent ICC(3,1) across all BMI categories, ranging from 0.873 to 0.932. The highest reliability was observed in participants with BMI < 40. All ICC values were statistically significant (*p* < 0.001), and the corresponding 95% confidence intervals indicated consistent reliability (Table 4).

The ROC analysis demonstrated that the 3MWT was a strong discriminator of individuals with low exercise tolerance, defined as 6MWT < 450 m. The AUC was 0.931, with an optimal 3MWT cut-off of 212 m, yielding sensitivity of 89.2% and specificity of 85.0% (Figure 1).

## 4. Discussion

The recent literature emphasizes the importance of functional capacity assessment in individuals with obesity, particularly those undergoing preoperative evaluation for bariatric surgery. The 6MWT has long been used to evaluate submaximal aerobic performance; however, its feasibility in individuals with severe obesity is increasingly being questioned due to factors such as premature fatigue, musculoskeletal discomfort, and spatial limitations in clinical settings [2,5,17]. As a result, shorter and more practical tests including the 2MWT, 3MWT, and 1MSTS have been proposed as alternatives. These tests have been applied in diverse populations, including older adults and individuals with cardiovascular or orthopedic conditions [8,18,19], but have not yet been directly compared as substitutes for the 6MWT in bariatric surgery candidates. Therefore, we sought to determine whether any of these short functional tests could serve as a reliable and valid substitute for the 6MWT in this specific population. The present study demonstrated that the 3MWT had the strongest association with 6MWT performance and yielded the highest test–retest reliability metrics among all tested measures. Moreover, the 3MWT demonstrated an excellent ability to discriminate individuals with reduced exercise tolerance (6MWT < 450 m). While the 2MWT also showed an acceptable correlation, the 1MSTS exhibited relatively weaker associations with the 6MWT, especially in participants with lower BMI values.

The 3MWT showed a robust and consistent correlation with the 6MWT, explaining a large portion of variance in the 6MWT distance, especially among individuals with BMI > 50. This aligns with evidence supporting the 3MWT as a valid proxy for submaximal aerobic capacity. For example, Beekman et al. highlighted how differences in course length significantly influence walk distance in COPD patients, underscoring the importance of standardized protocols in shorter tests [20]. Additionally, emerging data suggest that shorter walks like the 3 min test have high clinical sensitivity: a recent ERS abstract indicates 87–91% sensitivity in detecting oxygen desaturation, demonstrating similar prognostic potential to longer walks [21]. Compared to the 6MWT, the 3MWT is less time- and space-consuming, induces less fatigue, and is safer for individuals with severe obesity, who may experience joint pain or balance issues. Our cohort exhibited excellent test–retest reliability (ICC > 0.9), paralleling findings in the broader walking test literature [22]. Importantly, the ROC analysis in our study identified a cut-off of 212 m on the 3MWT, with strong sensitivity and specificity, which could serve as a critical threshold to detect submaximal aerobic impairment. Clinically, this would in in early prehabilitation decision-making for bariatric candidates who might not tolerate the 6MWT. However, it should be noted that the lower bound of the ICC confidence interval in participants aged over 50 fell within the moderate range, suggesting that the test’s reliability may be somewhat less consistent in this subgroup. Therefore, results in older patients should be interpreted with caution, and further age-stratified studies are warranted.

Although the 2MWT also demonstrated acceptable psychometric properties and a moderately strong association with the 6MWT, its clinical utility appears to be more nuanced than that of the 3MWT. Its shorter duration and minimal fatigue make it a practical option in time-restricted or resource-limited settings. Prior research has consistently supported the validity of the 2MWT across diverse clinical groups, including those with multiple sclerosis, spinal cord injury, and lower limb amputation [23,24,25,26]. However, emerging evidence suggests that the test may not adequately reflect sustained aerobic capacity, particularly in populations with obesity, where walking initiation is often cautious and performance may improve beyond the second minute [21]. This phenomenon may explain the relatively weaker correlation of the 2MWT in our bariatric cohort compared to the 3MWT. Moreover, shorter tests are more susceptible to pacing variability and motivational fluctuation, potentially limiting their discriminative precision [21,24,27]. Despite these limitations, the 2MWT remains clinically valuable in resource-constrained environments or when patients are unable to tolerate longer testing protocols. Its continued use is justified, especially when interpreted in conjunction with other markers of exertion or mobility.

In our study, the One-Minute Sit-to-Stand Test (1MSTS) demonstrated weak and inconsistent associations with the 6MWT, suggesting limited utility in estimating aerobic walking capacity among individuals with severe obesity. While the 1MSTS is increasingly recognized for its simplicity and safety, particularly in pulmonary and cardiac populations [28,29], its performance appears to rely more on lower-extremity muscle strength, balance control, and neuromuscular coordination than on sustained cardiovascular endurance [28,30,31]. This may partly explain the incongruity between 1MSTS repetitions and walking distance in our cohort, as obese individuals often display altered biomechanics, reduced postural transitions, and early fatigue during repeated sit-to-stand efforts [32]. These findings may explain the lack of agreement with walking-based assessments in our cohort, where excess body weight and joint loading could further compromise sit-to-stand mechanics. Therefore, while the 1MSTS may be useful in assessing overall functional mobility or strength-related limitations, its ability to detect aerobic impairment appears restricted—particularly in bariatric populations.

Some limitations of this study should be noted. Although the sample size was statistically adequate for correlation and regression analyses, the predominance of female participants (80.3%) may limit the generalizability of findings to male or more gender-balanced populations. Moreover, all participants were recruited from a single tertiary center, potentially reducing the external validity across broader clinical contexts. While the test procedures were standardized, individual factors such as fatigue, pain, sleep quality, or psychological status could not be fully controlled and may have influenced short-duration test performance. Despite these limitations, the strong correlations, high intraclass correlation coefficients, and excellent discriminative accuracy of the 3MWT support the robustness of our findings. Future studies employing longitudinal, multi-center designs with more diverse populations are recommended to confirm and expand upon these results.

## 5. Conclusions

This study demonstrated that, among the short functional tests examined, the 3MWT had the strongest validity, reliability, and discriminative accuracy in reflecting 6MWT performance in bariatric surgery candidates. It appears to be clinically useful alternative when standard walk testing is impractical. The 2MWT showed acceptable but comparatively lower alignment, making it a reasonable option in time-limited settings. The 1MSTS, while valuable for the assessment of functional mobility, exhibited weaker associations and may be less suitable for the estimation of aerobic walking capacity in this population. These findings support the use of the 3MWT in preoperative evaluation and highlight the need for further research to explore its longitudinal responsiveness and prognostic value.

## Figures and Tables

**Figure 1 healthcare-13-01883-f001:**
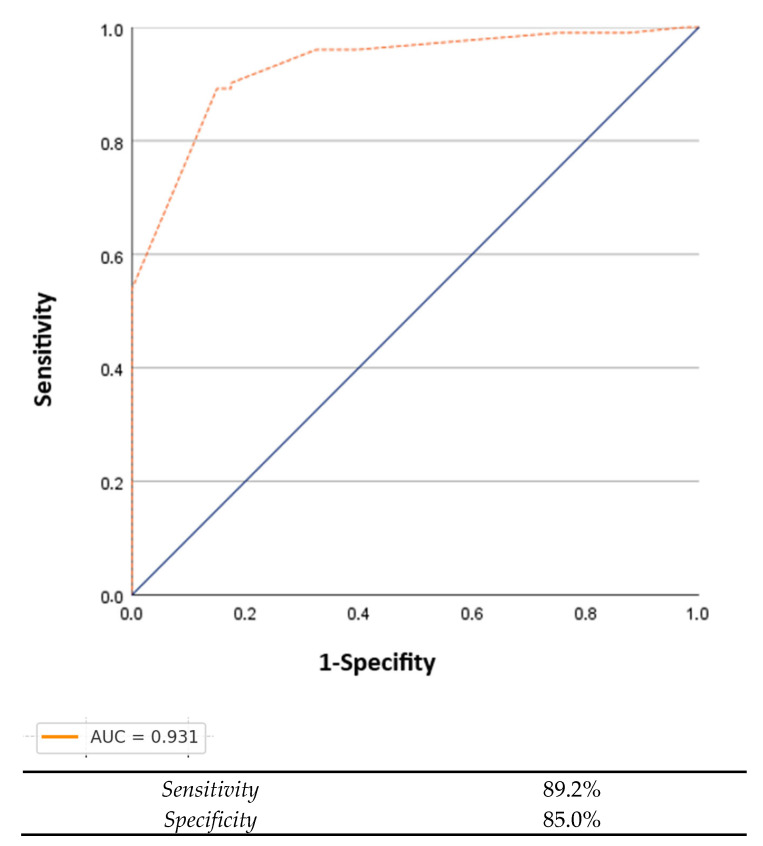
ROC curve of 3MWT for identification of low exercise tolerance (6MWT < 450 m).

**Table 1 healthcare-13-01883-t001:** Demographic characteristics of the participants.

Variable		Description *
Age		34.94 ± 10.61
BMI		45.92 ± 7.09
Waist-to-Hip Ratio		0.94 ± 0.08
Gender	*Male*	28 (19.7%)
	*Female*	114 (80.3%)
Smoking	*Yes*	49 (34.5%)
	*No*	93 (65.5%)
Six-Minute Walk Test		412.37 ± 81.11 m
Three-Minute Walk Test		206.72 ± 30.75 m
Two-Minute Walk Test		140.27 ± 20.02 m
One-Minute Sit-to-Stand Test		20.98 ± 4.14 repetitions

* Mean ± SD for continuous variables; frequency (%) for categorical variables.

**Table 2 healthcare-13-01883-t002:** Correlations between functional capacity tests and 6-Minute Walk Test across BMI categories.

Test	BMI < 40 *r* * (*p*)	BMI 40–50 *r* * (*p*)	BMI > 50 *r* * (*p*)
** *Three-Minute Walk Test* **	0.718 (0.0004)	0.765 (<0.001)	0.930 (<0.001)
** *Two-Minute Walk Test* **	0.645 (0.0021)	0.680 (<0.001)	0.756 (<0.001)
** *One-Minute Sit-to-Stand Test* **	0.172 (0.4697)	0.320 (0.0021)	0.339 (0.0576)

* Pearson correlation coefficients (r) and corresponding *p*-values are presented for each BMI category.

**Table 3 healthcare-13-01883-t003:** Linear regression between functional tests and 6MWT by BMI category.

Test	BMI < 40 *R*^2^ (*p*) *	BMI 40–50 *R*^2^ (*p*) *	BMI > 50 *R*^2^ (*p*) *
**3MWT**	0.515 (*p* = 0.001)	0.585 (*p* = 0.001)	0.865 (*p* = 0.001)
**2MWT**	0.416 (*p* = 0.002)	0.462 (*p* = 0.001)	0.572 (*p* = 0.001)

* R^2^ and *p*-values from linear regressions with 6MWT, by BMI category.

**Table 4 healthcare-13-01883-t004:** Test–retest reliability of the 3MWT by BMI category.

BMI Group	Mean ± SD (3MWT)	ICC *	95% CI (Lower–Upper)	*p*
** *<40* **	201.45 ± 6.22 m	0.932	0.742–0.977	<0.001
** *40–50* **	211.90 ± 3.21 m	0.900	0.840–0.936	<0.001
** *>50* **	209.22 ± 6.58 m	0.873	0.430–0.956	<0.001

* ICCs were based on a two-way mixed model with absolute agreement.

## Data Availability

The raw data supporting the conclusions of this article will be made available by the authors on reasonable request.

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
