# Peer review of "Less Time, Same Insight? Evaluating Short Functional Tests as Substitutes for the Six-Minute Walk Test and the Reliability and Validity of the 2MWT, 3MWT, and 1MSTS in Bariatric Surgery Candidates with Obesity"

_healthcare, 2025, doi:10.3390/healthcare13151883_

Round 1

Reviewer 1 Report

Comments and Suggestions for Authors

This cross-sectional study compared three physical performance tests, such as the Two-Minute Walk Test (2MWT), Three-Minute Walk Test (3MWT), and One-Minute Sit-to-Stand Test (1MSTS), with the Six-Minute Walk Test (6MWT) in the pre-operative assessment of 142 obese young (80% females) adults undergoing bariatric surgery. The 3MWT and, to a minor extent, but not the 1MSTS the 2MWT, correlated well with the 6WMT. The authors conclude the 3MWT may represent a valid alternative to the 6MWT, particularly in resource-limited settings.

GENERAL COMMENT

The study is per se of some interest, methodologically sound and in general well written. Its interest for the general readers of Healthcare is in my view questionable, whereas it might be better targeted to one of the numerous clinical journals in the field. There are some major issues that should be taken into account.

  1. The authors have recently published a study (DOI: 10.1007/s11695-025-07714-x) comparing the 6MWT with the Duke Activity Scale Index in obese patients, candidates for bariatric surgery. The rational of the study, the sample and the methods were quite similar to those of the present manuscript, to the point that the two studies might have been meaningfully combined into a single one. This appears as a not commendable example of “salami technique”.
  2. The authors used quite generously the term “predictive” (Abstract, Discussion). As a matter of fact, given the cross-sectional study design, there is no prediction here, but only associations.
  3. The study would be much more convincing if post-operative outcome measures (e.g., complications or functional capacity) were available.

Other minor points.

  1. Page 3, line 107. The authors wrote: “The distances covered at exactly 2 and 3 minutes were recorded separately using a stopwatch”. Distances are not measured with stopwatches. Please rephrase.

  1. Page 5, Figure 1. In a ROC curve plot, the x-axis represents 1-specifity, not specificity.

Author Response

Reviewer #1

1. The authors have recently published a study (DOI: 10.1007/s11695-025-07714-x) comparing the 6MWT with the Duke Activity Scale Index in obese patients, candidates for bariatric surgery. The rational of the study, the sample and the methods were quite similar to those of the present manuscript, to the point that the two studies might have been meaningfully combined into a single one. This appears as a not commendable example of “salami technique”.

Author’s Response: We sincerely thank the reviewer for this thoughtful and important observation. We respectfully clarify that although both studies were conducted within the same clinical setting, they address entirely different research questions, utilize different functional assessment tools, and most importantly are based on distinct participant samples.

  • The previously published study evaluated the reliability and validity of the Duke Activity Status Index as a self-reported alternative to the 6MWT.
  • The current study investigates the validity and clinical utility of three short performance-based field tests (2MWT, 3MWT, and 1-MSTS) as practical substitutes for the 6MWT.

Importantly, although both studies were conducted in the same bariatric surgery clinic, the sample populations are not the same. Our clinic serves a high volume of bariatric candidates, which enabled us to recruit entirely different groups of patients for each study. We have ensured that there is no participant overlap between the two studies.

2. The authors used quite generously the term “predictive” (Abstract, Discussion). As a matter of fact, given the cross-sectional study design, there is no prediction here, but only associations.

Author’s Response: We fully agree that the use of the term “predictive” in a cross-sectional design may lead to misinterpretation, as such a design does not support temporal causality or true prognostic inference. Accordingly, we have carefully revised the Abstract and Discussion sections to replace the term “predictive” with more appropriate alternatives such as “associative,” “discriminative,” or “correlational,” depending on the context.

3. The study would be much more convincing if post-operative outcome measures (e.g., complications or functional capacity) were available.

Author’s Response: We thank the reviewer for this valuable suggestion. While we fully agree that postoperative outcomes would enhance the clinical value of the study, our current design focused solely on preoperative functional assessment.

4. Page 3, line 107. The authors wrote: “The distances covered at exactly 2 and 3 minutes were recorded separately using a stopwatch”. Distances are not measured with stopwatches. Please rephrase.

Author’s Response:  Thank you for highlighting this phrasing issue. We agree with the reviewer and have reworded the sentence to clarify that the stopwatch was used for timing, not for measuring distance. The revised sentence now reads: “Distances covered at exactly 2 and 3 minutes were recorded, while time was tracked using a stopwatch.”

5. Page 5, Figure 1. In a ROC curve plot, the x-axis represents 1-specifity, not specificity.

Author’s Response:  We thank the reviewer for this correction. The x-axis label in Figure 1 has been revised from “Specificity” to “1 – Specificity” in accordance with standard ROC plotting conventions.

Reviewer 2 Report

Comments and Suggestions for Authors

- Your title does not reflect the scope of your study or your population. My recommendation is "Reliability, validity, and feasibility of the Two-Minute Walk Test, Three-Minute Walk Test, and Sit-to-Stand Test in patients with ..."

- It says you also evaluated reliability in the statistical analysis, but you didn't specify it in the objectives section. Also, how did you evaluate feasibility? I didn't see it in the methods section.

- What do you mean by "clinical utility"? is it "reliability"?

- What statistical tool did you use to calculate sample size? MCID values are used in longitudinal studies, in comparing outcome measures across multiple groups, or in experimental studies. Reliability and validity studies require different statistical methods, some of which are described in the literature. Please recalculate your sample size using these methods. Your sample size will likely be sufficient, but your method is incorrect (Walter et al. SAMPLE SIZE AND OPTIMAL DESIGNS FOR RELIABILITY STUDIES; Bujang et al. Sample Size Guideline for Correlation Analysis).

- Not conducting tests randomly may create bias due to the learning and fatigue effects in performance-based tests. Please note this as a limitation.

- Why did you only look at the reliability of the 3-MWT test and why did you separate it into age groups? Also, which model and type did you use for the ICC (ICC(3,k) or (2,k)? Please see the reference (Koo and Li. A Guideline of Selecting and Reporting Intraclass Correlation Coefficients for Reliability Research).

- When interpreting the ICC, interpretation is based on 95% CI intervals. So, based on your results, the reliability of the 3-MWT test is moderate to excellent in the first ICC step and poor to excellent in the last. This means it is not a reliable test for people over 50. Please review the reference above and other ICC interpretation guidelines. Please base your discussion accordingly.

Comments on the Quality of English Language

Good.

Author Response

Reviewer #2

1. Your title does not reflect the scope of your study or your population. My recommendation is "Reliability, validity, and feasibility of the Two-Minute Walk Test, Three-Minute Walk Test, and Sit-to-Stand Test in patients with ..."

Author’s Response:  We sincerely thank the reviewer for this constructive recommendation. We appreciate the importance of clarity in reflecting the study’s scope and population. To preserve the originality of the current title while incorporating your suggestion, we have revised the title as follows:

Less Time, Same Insight? Evaluating Short Functional Tests as Substitutes for the Six Minute Walk Test Reliability and Validity of the 2MWT, 3MWT, and 1MSTS in Bariatric Surgery Candidates with Obesity

2. It says you also evaluated reliability in the statistical analysis, but you didn't specify it in the objectives section. Also, how did you evaluate feasibility? I didn't see it in the methods section.

Author’s Response:  We have updated the Objectives section to explicitly state that the study includes the evaluation of test–retest reliability using ICC values. In addition, to avoid confusion, all references to “feasibility” have been removed from the title and manuscript text, as feasibility was not directly assessed through predefined criteria.

3. What do you mean by "clinical utility"? is it "reliability"?

Author’s Response:  In our manuscript, “clinical utility” was intended to refer to the overall practical applicability of each test in bariatric clinical settings.

4. What statistical tool did you use to calculate sample size? MCID values are used in longitudinal studies, in comparing outcome measures across multiple groups, or in experimental studies. Reliability and validity studies require different statistical methods, some of which are described in the literature. Please recalculate your sample size using these methods. Your sample size will likely be sufficient, but your method is incorrect (Walter et al. SAMPLE SIZE AND OPTIMAL DESIGNS FOR RELIABILITY STUDIES; Bujang et al. Sample Size Guideline for Correlation Analysis).

Author’s Response:  We acknowledge that our initial sample size estimation based on MCID was not appropriate for a correlation-based validity study. As suggested, we recalculated the required sample size using the guidelines provided by Bujang et al. (2016), which are specific to correlation analyses. Based on existing literature, we anticipated a minimum correlation of r = 0.30 between the short functional tests and the 6MWT. This reflects a moderate association and aligns with previous studies in similar populations. Assuming α = 0.05 and a power of 80%, the recommended sample size is 85 participants. Our actual sample size of 142 participants exceeds this requirement, confirming the adequacy of our sample. The Methods section has been revised accordingly, and MCID-based justification has been removed.

5. Not conducting tests randomly may create bias due to the learning and fatigue effects in performance-based tests. Please note this as a limitation.

Author’s Response:  To minimize potential learning and fatigue effects, we indeed conducted the performance-based tests in a randomized order for each participant. This procedural detail has now been clarified in the Methods section.

6. Why did you only look at the reliability of the 3-MWT test and why did you separate it into age groups? Also, which model and type did you use for the ICC (ICC(3,k) or (2,k)? Please see the reference (Koo and Li. A Guideline of Selecting and Reporting Intraclass Correlation Coefficients for Reliability Research).

Author’s Response:  Although test–retest measurements were collected for all three functional tests, we focused on reporting the reliability of the 3-MWT, as it demonstrated the highest validity and discriminative performance in our analyses. Given its promising clinical utility, we chose to highlight its reliability outcomes in the manuscript. Regarding ICC analysis, we followed the guidelines of Koo and Li (2016) and used a two-way mixed-effects model, single rater, absolute agreement type, corresponding to ICC(3,1).

The analysis was also stratified by age group (≤50 vs. >50 years) to explore whether age-related physiological differences may affect test consistency, which is particularly relevant in bariatric populations.

7. When interpreting the ICC, interpretation is based on 95% CI intervals. So, based on your results, the reliability of the 3-MWT test is moderate to excellent in the first ICC step and poor to excellent in the last. This means it is not a reliable test for people over 50. Please review the reference above and other ICC interpretation guidelines. Please base your discussion accordingly.

Author’s Response: We agree that interpretation of ICC values should always be made in conjunction with their 95% confidence intervals, as emphasized by Koo and Li (2016) and other guidelines. As the lower bound of the ICC confidence interval in the >50 age group fell within the poor-to-moderate range, we acknowledge that the reliability of the 3-MWT in this subgroup may be less robust than in younger participants. Accordingly, we have revised the Discussion section to reflect this limitation and avoid overstatement.

Round 2

Reviewer 1 Report

Comments and Suggestions for Authors

The authors answered satisfactorily to my minor remarks. 

As far as my main objection of an excessive overlap with the previously published study, although the research questions are different, they fall quite within the same area. One of the main variables (6MWT) was exactly the same in the two studies. The enrollment time window was not reported, neither in the previous investigation nor in the present one: therefore, the authors statement that the samples were different cannot unfortunately be verified.  

Reviewer 2 Report

Comments and Suggestions for Authors

The authors implemented most of the editor and reviewer suggestions. All concerns raised have been addressed and resolved.

The article has been improved over the original version.

No further questions or comments.